# Enriched sleep environments lengthen lemur sleep duration

**Alexander Q. Vining** [1,2,3]*, **Charles L. Nunn**[4,5], **David R. Samson**[4,6]*

**1** Animal Behavior Graduate Group, University of California, Davis, Davis, California, United States of America, **2** Department for the Ecology of Animal Societies, Max Planck Institute of Animal Behavior, Radolfzell, Germany, **3** Department of Biology, University of Konstanz, Konstanz, Germany, **4** Department of Evolutionary Anthropology, Duke University, Durham, North Carolina, United States of America, **5** Duke Global Health Institute, Duke University, Durham, North Carolina, United States of America, **6** Department of Anthropology, University of Toronto, Mississauga, Canada

* avining@ab.mpg.de (AQV); david.samson@utoronto.ca (DRS)

**Data Availability Statement:** All data used in this study are publicly available on the project github repository https://github.com/aqvining/Lemur_Sleep_Site_Enrichment.

**Funding:** This research was supported by a grant from the Natural Sciences and Engineering

## Abstract

Characteristics of the sleep-site are thought to influence the quality and duration of primate sleep, yet only a handful of studies have investigated these links experimentally. Using actigraphy and infrared videography, we quantified sleep in four lemur species (*Eulemur coronatus*, *Lemur catta*, *Propithecus coquereli*, and *Varecia rubra*) under two different experimental conditions at the Duke Lemur Center (DLC) in Durham, NC, USA. Individuals from each species underwent three weeks of simultaneous testing to investigate the hypothesis that comfort level of the sleep-site influences sleep. We obtained baseline data on normal sleep, and then, in a pair-wise study design, we compared the daily sleep times, inter-daily activity stability, and intra-daily activity variability of individuals in simultaneous experiments of sleep-site enrichment and sleep-site impoverishment. Over 164 24-hour periods from 8 individuals (2 of each species), we found evidence that enriched sleep-sites increased daily sleep times of lemurs, with an average increase of thirty-two minutes. The effect of sleep-site impoverishment was small and not statistically significant. Though our experimental manipulations altered inter-daily stability and intra-daily variability in activity patterns relative to baseline, the changes did not differ significantly between enriched and impoverished conditions. We conclude that properties of a sleep-site enhancing softness or insulation, more than the factors of surface area or stability, influence lemur sleep, with implications regarding the importance of nest building in primate evolution and the welfare and management of captive lemurs.

## Introduction

Sleep is a period of behavioral quiescence and reduced responsiveness to external stimuli, thus making it a vulnerable and dangerous state [1]. Consistent with this observation, comparative studies have revealed that risk of predation at the sleep-site covaries negatively with sleep duration across mammals [2–4]. Sleep-sites likely also vary in quality and security along dimensions that involve the physical comfort of the substrate, level of concealment from conspecifics

Research Council of Canada (https://www.nserc-crsng.gc.ca/index_eng.asp; RGPIN-2020-05942) awarded to author DRN. The funders had no role in study design, data collection and analysis, decision to publish, or preparation of the manuscript.

**Competing interests:** The authors have declared that no competing interests exist.

and predators, and thermal properties [5–7]. A major question concerns how characteristics of the sleep site influence sleep quality, and how that in turn influences organismal function and fitness.

Considerable effort has been put into studying the role of sleep-site characteristics for apes. The *sleep quality hypothesis*, for example, proposes that apes construct beds to permit less fragmented, undisturbed sleep that promotes sleep quality through either greater sleep intensity [deeper slow-wave sleep (SWS) and/or REM sleep] or longer individual sleep stages [8–11]. This hypothesis has been supported by increased effort chimpazees put into building more complex nests [12] and observations of orang-utans (*Pongo pygmaeus wurmbii*) in Southern Borneo selecting sleep sites for comfort and stability rather than predator defence [13]. More recently, comparative analysis has shown that the sleep of captive orang-utans (*Pongo spp*.) is less frequently interrupted by movements of the head and body than the sleep of baboons (*Papio papio*.) [14]. Moreover, experimental work has demonstrated that orang-utans exhibit higher quality sleep (defined by less gross-motor movement and greater overall sleep times) when using complex sleeping platforms [15]. For wild, individually sleeping apes, sleep-site modifications may improve sleep through several mechanisms. For one, the removal or covering of protruding substrate reduces stress on tissues [12]. The relaxed skeletal muscle tone of REM sleep also puts branch sleeping animals at risk of falling; enlarged surface area and functional concavity of nests likely reduce this risk and enable greater quantities of REM sleep.

Other primates exhibit a wide range of sleep-site selection behaviors, but do not build daily nests as do great apes [16]. Guinea baboons (*Papio papio*), for example, will sleep on both cliffs and tree branches, often on terminal branches of emergent trees. Many New World monkeys such as the golden lion tamarin (*Leontopithecus rosalia*) often sleep in tree cavities, while one population of tufted capuchin monkeys (*Cebus apella*) has shown a proclivity for sleeping in the leaves of *Jessenia* palms. Among lemurs, variation is sleep-site behavior is as broad as in other primates. Some lemurs, like great apes, make nests (*Mirza coquereli*, *Microcebus myoxinus*, *Galagoides demidoff)*, many sleep in tree holes, and yet more are branch sleepers [6].

Investigation of how sleep-sites affect lemur sleep is interesting for at least three reasons: 1) the use of nests by some lemurs and by great apes, but not Haplorrhine monkeys, begs the question of why (and when) this behavior has appeared and disappeared in the primate lineage, 2) variation amongst lemur species creates opportunities to study the ecological factors that select for different sleep phenotypes, and 3) reconstructing the sleep phenotype of ancestral lemurs will help us understand the starting point, and subsequent constraints, of evolution in primate sleep.

The first question has been partly answered by observing differences in the nesting behavior of lemurs and apes: lemurs do not conduct nightly sleep site modifications as great apes do–making their nests functionally more akin to those of birds than apes [6]. Cladistic differences in sleep-site modification behavior are also likely explained in part by advanced cognition and social learning in apes [5, 17]. An important aspect of understanding how and why sleep phenotypes vary across primates is determining the role that sleep-site comfort has on the sleep quality of non-nest sleeping primates. If aspects of nests, such as comfort, shelter, and stability improve sleep quality, it would suggest non-nesting species face greater costs of nest building, perhaps from the need to defend nests or the inability to change sleeping locations.

In previous work, we found that disrupted sleep influences aspects of lemur behavior and cognition [18]. In wild grey mouse lemurs, similar cognitive tests predicted body condition and survival [19], suggesting that sleep-related changes have functional consequences. Here, we investigate the effect of sleep-site characteristics on sleep-wake activity and duration in four species of lemur (*Eulemur coronatus*, *Lemur catta*, *Propithecus coquereli*, and *Varecia rubra*) by experimentally enriching and impoverishing sleep-sites in pairs of captive lemurs

simultaneously. We experimentally tested the hypothesis that the comfort and stability of the sleep-site influences lemur sleep quality. Based on this hypothesis, we predicted that 1) enriching a sleep-site with soft, insulating materials would increase sleep duration and decrease sleep fragmentation (measured through intra-daily variability in activity patterns) and 2) impoverishing a sleep-site by removing flat, stable, above-ground surfaces would decrease total sleep time and increase sleep fragmentation. Thus, in addition to implications for animal welfare, understanding the links between sleep sites and sleep may also inform understanding of primate behavior, ecology, and evolution.

## Methods

### Study subjects

Research was performed at the Duke Lemur Center (DLC), in Durham, North Carolina, USA, where subjects were housed in dyadic, sex-balanced groups (i.e., with one male and one female). We generated actigraphic data from eight individuals, with one male and one female from each species. For detailed biographical information on the study subjects, see Bray et al. [20]. Animals received unlimited access to water and fresh fruit, vegetables, and Purina monkey chow on a daily basis. Animal use and methods were approved by the Duke University Institutional Animal Care and Use committee (Protocol #: A236-13-09) and the DLC Research Committee.

### Data collection

The study was conducted over four months from April to July in 2016. Given each experimental protocol involved the work of multiple DLC staff and researchers, the research needed to be distributed on a per species basis as follows: *Varecia*: April 6 –April 26, *Lemur*: May 26 –June 15, *Propithecus*: Tuesday June 7 –Monday June 27, *Eulemur*: July 14 –August 3. This effort generated a dataset of 40–42 nights per species, totaling 164 twenty-four-hour periods, with circadian activity continuously recorded using MotionWatch 8 (CamNtech) tri-axial accelerometers. These actigraphic sensors are lightweight (7 g) and attached to standard nylon pet collars. Animals were monitored by DLC caretaking staff closely for two hours after collaring and at regular intervals throughout the study to ensure no adverse reactions to the collar; there were no reports of abnormal behavior beyond some scratching and head-shaking immediately following the collaring. The study took place in indoor housing to control for temperature and light conditions.

Using actigraphy data, response variables were generated from processed activity logs recorded at one-minute epochs. Recent advances in scoring algorithms have increased accuracy in detecting wake-sleep states and total sleep times [21]. We generated *twenty-four-hour total sleep times* (TST) for individuals in each species. We followed protocols used in previous primate sleep studies [22–26] and in prior work by our group performed at the DLC [20, 27]. The sensor sampled movement once per second at 50 Hz and assigned an activity value, referred to as "counts", per one-minute epoch. Previous studies have used the operational definition of behavioral sleep measured via actigraphy as the absence of any force in any direction during the measuring period [28]. We similarly determined that animals were consistently at rest, and thus inferred sleep states (i.e., sustained quiescence in a species-specific posture), when actigraphy count values were equal to zero. We selected this criteria as the most conservative measure of sleep available to us because CamNtech does not make available its algorithm for determining count values from acceleration data (but see Van Hees et al. [29] for an approximation of a similar proprietary actigraphy algorithm). Following recommendations for validating actigraphy-based inferences of sleep state [30], a previous study compared

infrared videography of sleeping lemurs to actigraphy counts from the same model of collar used in the current study. Lemurs could be clearly seen in the videos to make minor movements such as looking around or adjusting body position during 1 minute epochs with four or more activity counts, but such motions were not detected during epochs with fewer than four activity counts [27]. Nonetheless, using motion as a proxy for sleep leaves open the possibility that we may detect periods of relaxed wakefulness as sleep, and thus requires appropriate care when interpreting the results.

## Experimental procedure

In a pair-wise experimental design, individuals from each species underwent two weeks of simultaneous testing after being measured in a baseline condition. During week one (baseline), each pair experienced normal sleep conditions. These conditions were the same as normal indoor operating protocol for the DLC, where individuals had access to two housing cells with mounted, raised shelves and basic amenities (e.g., raised square crates attached to the walls). To reduce the confounds of outdoor lighting and temperature, the baseline conditions only differed from normal conditions in that individuals were restricted to sleeping indoors, where temperatures were maintained within a few degrees of 78˚F and the most substantial source of light came from dim emergency lighting at the ends of enclosure hallways. Baseline conditions differed from the experimental conditions in the options for sleep, with animals allowed to sleep socially (in pairs) during baseline but separated at night in the experiments. Though this presents a potential confound (with regard to the effect of sleep substrates) in comparisons of baseline conditions to experimental conditions, this decision reduced the overall impact of our study on the lemurs and we control for this confound by contrasting sleep times in our two experimental conditions, in which social conditions were the same.

During week two–the beginning of the pair-wise experiment–an individual was randomly chosen for one of two experimental conditions: *sleep enrichment* or *sleep impoverishment*. The other individual of the pair underwent the opposite treatment. One treatment was applied to each of the two housing cells the lemur pair had access to during baseline, and subjects restricted to their respective housing cell following evening caretaking. The third week was a reversal of the previous week's condition, with individuals switching sleep-sites and thus experiencing the opposite condition. During experimental nights, individuals were isolated from the pair-mate, but they were in vocal communication and could touch one another between the enclosures, thus providing some of the typical social conditions they experienced in the baseline period. This was done to reduce the influence of separation on individual level sleep-wake expression.

Housing cells were enriched by providing lemurs with a high-quality plastic sleeping box (30 cm x 60 cm) with open slats on the side to permit ventilation; additionally, a 2.5 cm slab memory foam mattress was embedded in the base of the box with a small nylon blanket placed on the top of the mattress (Fig 1). Housing cells were impoverished by removing all enrichments items (including sleeping crates available in baseline conditions) and removing normally available wall shelves, leaving only narrow ledges for above ground perching during sleep.

We used infrared videography to determine whether the subject in the sleep enrichment condition was using the sleeping box. On only one night did an individual lemur (Beatrice, *P. coquereli*) fail to use the enriched sleeping crate when available. In all conditions, the nighttime period was considered to start at 18:00 and end at 06:00. During the day, enclosures were returned to baseline conditions and the animals were allowed to move freely between cells.

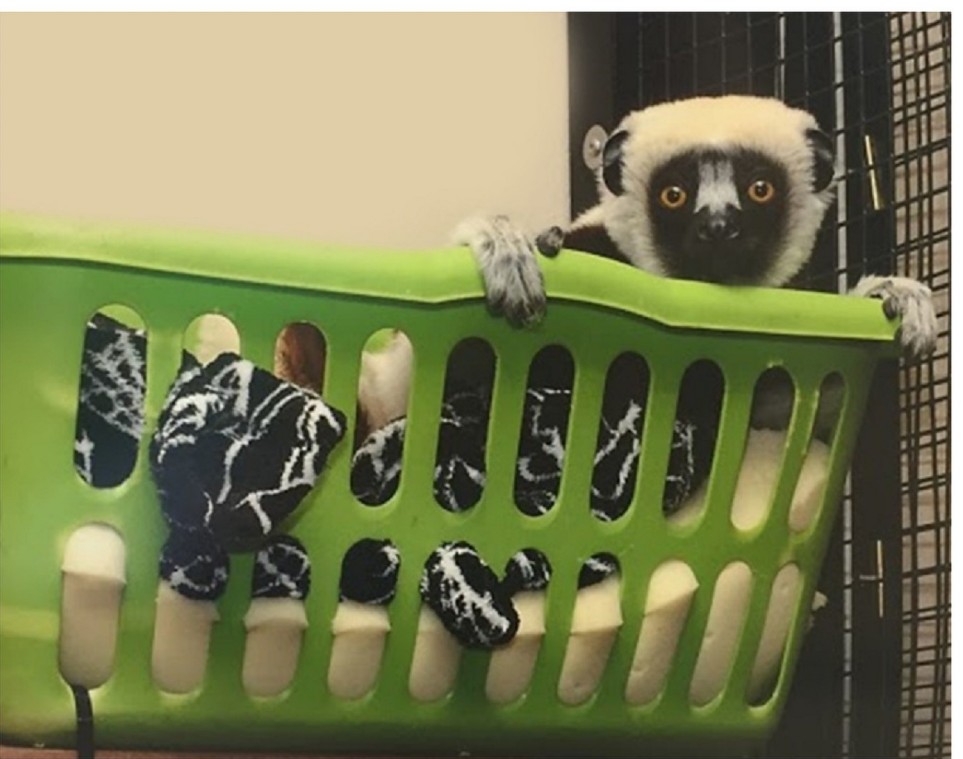

**Fig 1. The sleep enrichment experimental condition.** A subject *Propithecus coquereli* perched in the enriched sleep-site provided in the sleep enrichment experimental condition.

## Data analysis

We generated descriptive statistics characterizing the distribution of inferred sleep time throughout the 24-hour period by individuals and across the different experimental conditions. To test whether our experimental manipulations of sleep-site condition had a meaningful effect on lemur sleep times, we followed the approach of Pinhiero & Bates [31], building and testing a series of nested linear mixed-effects models. We began by modeling only the random effect of individual (nested within species), thus establishing a reference model that acknowledges sleep times are individually variable but assumes our experimental manipulations had no effect (Model 0). Because sleep is a biorhythm and likely to contain temporal dependencies, we expanded the reference model to include within-subject temporal autocorrelation in TST across 24-hour periods. Based on the autocorrelation function of our sleep data, we compared Model 0 to a first order autoregressive model of TST within subjects (Model 1). Finally, we tested the null hypothesis that our experimental manipulations had no effect on TST by adding to Model 1 parameters describing our experimental structure. These parameters were the coefficients for the three levels of experimental condition, the two orders in which the conditions were presented (used to control for potential order effects [32]), and their interactions (Model 2), resulting in the equation

$$
\begin{aligned}
TST_{j,s,t} = \mu + {} & \phi_1 TST_{j,s,t-1} + \beta_C Condition_{j,t} + \beta_O Order_j + \beta_{C,O} Condition_{j,t} * Order_j + U_s + V_j \\
& + E_{j,s,t}
\end{aligned}
$$

where $TST_{j,s,t}$ is the TST of individual $j$ of species $s$ on day $t$, $\mu$ is the intercept (the baseline TST for subjects presented with the enrichment condition prior to the impoverishment condition),

$\phi_1$ is the magnitude of the first order temporal auto-regression, $\beta_C$ is the regression coefficient for the experimental condition given by $Condition_{j,t}$, $\beta_O$ is the regression coefficient for the order of experimental conditions (enrichment first versus impoverishment first) given by $Order_j$, $\beta_{C,O}$ is the regression coefficient for the interaction of the condition-order pair given by $Condition_{j,t}*Order_j$, $U_s$ is a random intercept for species $s$, $V_j$ is a random intercept for individual $j$, $E_{s,j,t}$ is an error term, and the latter three terms are assumed to follow normal distributions with mean 0 and variances $\sigma_U^2$, $\sigma_V^2$, and $\sigma_E^2$, respectively.

We used delta AICs at each step to assess whether the more complex model provided a sufficiently improved fit to the data. Before making inferences about the effects of our experimental structure, we plotted the normalized residuals of Model 2 against 1) predicted values and 2) the quantiles of a standard normal distribution to conduct diagnostic checks of our model assumptions. Concluding that Model 2 met all necessary assumptions, we calculated the contrasts of each level of experimental condition (including baseline and marginal to order) and tested for significant differences in TST between the baseline condition and the two experimental conditions using ANOVA. Finally, we calculated the intra-class correlation coefficients (ICCs) to compare the proportion of unstructured variance in our data and used this comparison to determine when our limited sample size prevented us from adding additional complexity to our statistical model. We fit all models to our data using the function lme of the library nlme v3.1 [33] in R version 4.0.4 [34]. Our analysis can be fully reproduced using code and documentation reported in the S1 File.

**Inter-daily stabililty and intra-daily variability.** In addition to testing for changes in total sleep time, we also calculated from the raw actigraphy data two metrics of sleep disruption first described in Van Someren et al. [35]: inter-daily stability (IS) and intra-daily variability (IV). Inter-daily stability is defined as the ratio of 1) the variance of averaged actigraphy counts across epochs around the grand mean to 2) the overall variance. In other words, it measures how consistent actigraphy measures are for each minute in a day across days, and thus gives an indication of how well activity levels are entrained to dial rhythms. Intra-daily variability is the ratio of 1) the squares of the difference in counts between all successive minutes and 2) the mean square differences in counts relative to the grand mean; it gives an indication of how fragmented activity patterns are within a given experimental period.

We calculated both metrics within each experimental condition for all subjects. We then analyzed these descriptive measures much like TST, fitting a linear model to each using fixed effects for experimental condition and order as well as their interaction and random effects for individual and species (nested). As with TST, we contrasted the effects of each experimental condition after marginalizing over the effects of order and its interaction with experimental condition.

## Results

Individual sleep times are presented in Fig 2. TST, inter-daily stability, and intra-daily variability are summarized by condition in Table 1. AIC revealed that Model 2 is the preferred model for TST (Model 2 AIC = 1742; Model 1 ΔAIC = 12.4; Model 0 ΔAIC = 16.6). The first order autoregressive effect in this model is small ($\hat{\varphi}_1 = 0.09$, 95% CI [-0.08, 0.26]). Estimated contrasts of TST between experimental conditions are presented in Fig 3. The estimated contrast of TST between the enriched condition and the baseline condition (marginal to order) is 32.0 minutes with a standard error of 8.72 and the hypothesis of no difference between enriched and baseline can be ruled out (*Enriched—Baseline*: df = 152, F-value = 13.0, p < 0.001). The contrast between the impoverished and baseline condition is not significantly different from 0 (*Impoverished—Baseline*: mean = -0.18 minutes, SE = 8.72, df = 152, F-value = <0.001

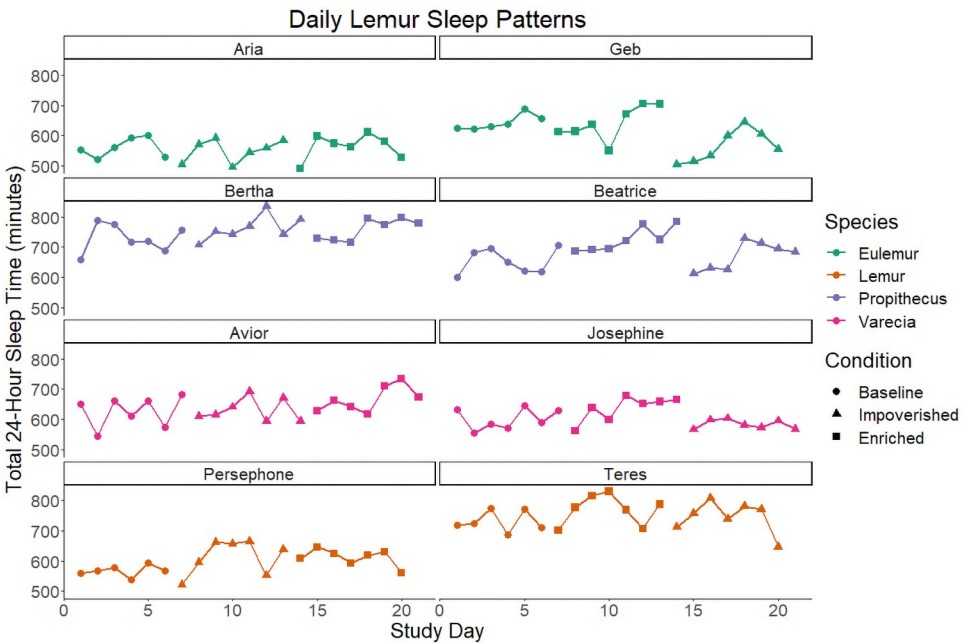

**Fig 2. Daily lemur sleep patterns.** Total sleep time in minutes for each observed 24-hour period, plotted by individual across days. Lemurs exposed to the impoverished condition first are presented on the left, those exposed to the enriched condition first on the right.

p = 0.984). The majority of unstructured variance in TST given Model 2 is attributable to individual differences ($ICC_{subject} = 0.592$), most of the remaining variance to the error term ($ICC_{Error} = 0.301$), and only a small proportion to species level differences ($ICC_{Species} = 0.107$).

Linear mixed-effect models of Inter-Daily Stability and Intra-Daily variability both produced normally distributed residuals; the estimated effects of experimental conditions on both metrics are presented in Fig 4. Both the enriched condition and the impoverished condition yielded higher inter-daily stability in sleep than the baseline condition, though the effects were small and non-significant (Enriched v. Baseline: mean difference = 0.013, se = 0.021, df = 12, F-Value = 0.38, p = 0.55; Impoverished v. Baseline: mean difference = 0.021, se = 0.021,

**Table 1. Summarized sleep times by individual and experimental condition.**

| ID | Sex | Species | Baseline | | | Enrichment | | | Impoverishment | | |
|---|---|---|---|---|---|---|---|---|---|---|---|
| | | | Mean TST | IS | IV | Mean TST | IS | IV | Mean TST | IS | IV |
| **Aria** | F | *E. coronatus* | 560 | 0.36 | 0.77 | 563 | 0.45 | 0.67 | 550 | 0.41 | 0.67 |
| **Geb** | M | *E. coronatus* | 643 | 0.39 | 0.74 | 642 | 0.42 | 0.64 | 565 | 0.51 | 0.56 |
| **Bertha** | F | *P. coquereli* | 729 | 0.32 | 0.85 | 759 | 0.40 | 0.76 | 763 | 0.40 | 0.74 |
| **Beatrice** | M | *P. coquereli* | 654 | 0.32 | 0.99 | 726 | 0.38 | 0.85 | 670 | 0.39 | 0.87 |
| **Avior** | M | *V. rubra* | 626 | 0.38 | 0.67 | 666 | 0.31 | 0.72 | 631 | 0.32 | 0.55 |
| **Josephine** | F | *V. rubra* | 601 | 0.31 | 0.83 | 636 | 0.27 | 0.90 | 583 | 0.29 | 0.94 |
| **Persephone** | F | *L. catta* | 568 | 0.34 | 0.80 | 611 | 0.31 | 0.71 | 613 | 0.29 | 0.79 |
| **Teres** | M | *L. catta* | 731 | 0.28 | 0.97 | 769 | 0.28 | 0.82 | 745 | 0.27 | 0.71 |

Mean sleep times (TST) per 24-hour period (minutes), inter-daily stability, and intra-daily variability for each study individual across conditions. All experimental periods were seven days, except Aria, Geb, Persephone, and Teres in the Baseline condition, which were six days.

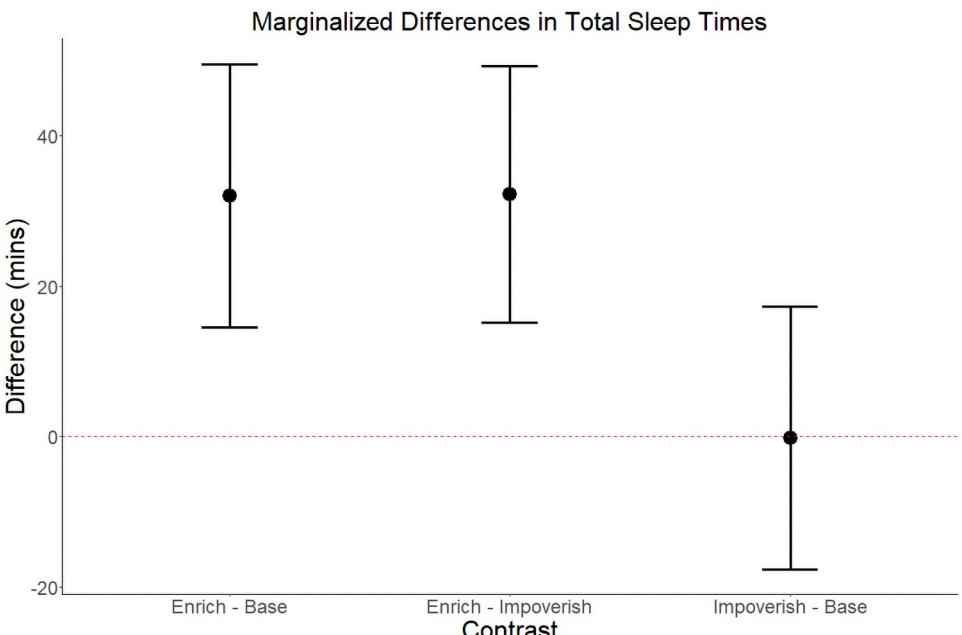

**Fig 3. Estimated effects of experimental condition on total sleep time.** The estimated differences in total sleep time (in minutes and marginalized over the estimated effects of condition order) for each pair of experimental conditions. Error bars represent two standard errors of the estimated population mean difference for the given contrast.

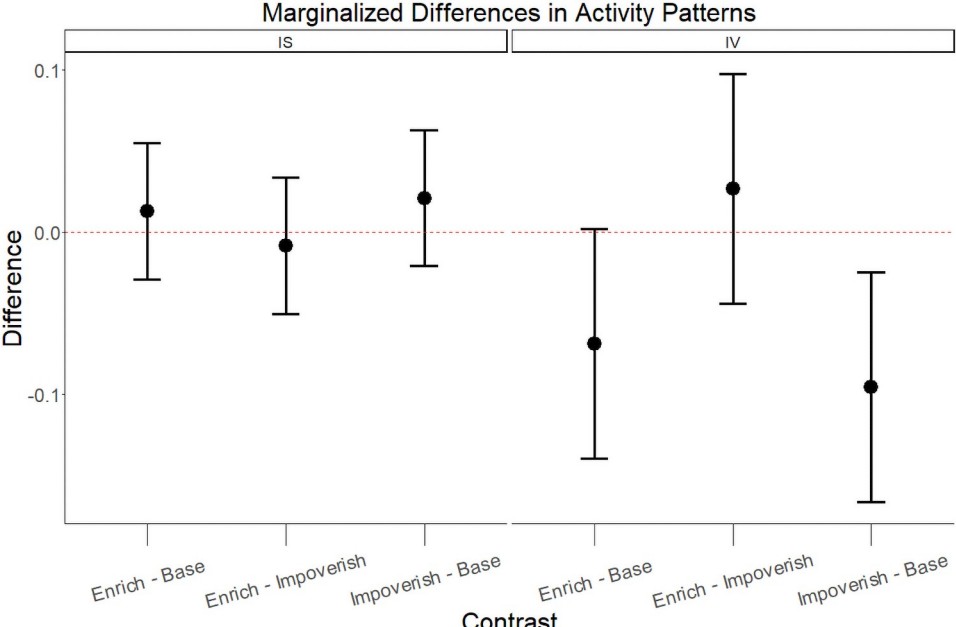

**Fig 4. Marginalized differences in activity patterns between experimental conditions.** The estimated mean difference in Inter-Daily Stability (IS) and Intra-Daily Variability (IV) of lemur activity patterns between experimental condition, marginalized over estimated effects of experimental order. Error bars represent two standard errors of the estimated population mean difference for the given contrast.

df = 12, F-Value = 1.02, p = 0.33). Both the enriched and impoverished conditions yielded decreased intra-daily variability relative to baseline conditions, where the effect was noteworthy for the enrichment condition and statistically significant in the impoverishment condition (Enriched v. Baseline: mean difference = -0.069, se = 0.035, df = 12, F-Value = 3.75, p = 0.08; Impoverished v. Baseline: mean difference = -0.095, se = 0.035, df = 12, F-Value = 7.24, p = 0.02).

## Discussion

The results of our study provide evidence that, for a limited subset of lemur species that do not nest outside the context of infant care, provisioning sleep sites with soft and insulated materials lengthens daily sleep durations. Among the models that we compared, the one that explicitly accounted for experimental manipulations was the most likely given our data. More importantly, we can conclude with confidence from this model that sleep site enrichment adds, on average, between 23 and 41 minutes of daily sleep relative to the baseline conditions (these values are the marginal contrasts, which account for any effects of the order experimental conditions were presented in). These conclusions are robust to sample size, but the small number of individuals in our study prevents us from making further conclusions about whether the results were driven by particularly strong effects in any subset of age, sex, or species classifications. An additional caveat is that we measure body-motion, rather than brain activity, as a proxy for sleep; it is possible we are measuring changes in wakeful resting states and we make no conclusions regarding specific sleep states such as REM or slow-wave sleep.

Interestingly, the manipulations of our impoverishment condition had no discernable impact on daily sleep times of our lemur subjects. Individuals in this condition were typically observed sleeping on narrow window ledges in their enclosures rather than on the ground. Though we cannot say for certain that the surface area–and hence stability–of lemur sleep sites do not affect sleep (we simply fail to reject this hypothesis), the differing impact of our impoverishment and enrichment conditions allows for some important inferences. First, we infer that the presence of soft, insulating materials is *more* important to our subjects' sleep duration than the size of the sleeping substrate. Second, it allows us to treat the impoverishment condition as a control, indicating that incidental effects common to both manipulations, such as the separation of lemur pairs at night or the disruption of typical caretaking routines, are not sufficient to explain the differences between the enriched and impoverished conditions. Such confounds, however, may alter the differences of both conditions from baseline; thus, caution is warranted in determining whether it was the enriched or impoverished condition that had the greater effect on sleep time.

Measures of inter-daily stability and intra-daily variability did not differ between experimental conditions. Intra-daily variability in activity patterns did, however, decrease substantially in both experimental conditions relative to the baseline. This suggests a confounding aspect of our experimental manipulations resulted in less fragmented sleep; the change in social sleep conditions (from paired to isolated sleeping) offers a logical explanation. Some caution should be taken in interpreting the reduction in intra-daily variability as a consequence of less fragmented sleep *per se*, as this is a measure of general activity patterns and could also be affected by fragmentation in other activities such as play or alert rest. Regardless of what caused changes in activity between baseline and experimental conditions, we failed to find any significant difference in intra-daily variability or inter-daily stability between the enriched and impoverished conditions.

Given that lemurs in our study experienced overall greater sleep duration in the enriched sleeping condition, it may have been that the sleep architecture of ancestral fixed-point

sleeping primates was deep and high quality, whereas later larger bodied primates that were branch sleepers traded longer sleep periods for other advantages. There are many potential trade-offs that could result in the use of less comfortable sleep sites, despite the evidence we found that such sites could reduce total sleep times. These include predator defense, thermo-regulation, parasite avoidance, group cohesion, and resource distribution [5, 36]. The role of predation may be particularly important for lemurs, all of which are highly predated by raptors, boas, and fossa [37]. The high number of cathemeral (active both at night and during the day) lemur species is hypothesized to result from the similarly 24-hour threat of these predator species [38]. High predation threat may also help explain why lemurs, but not other Malagasy mammals, show marked increases in female to male body size ratios relative to their mainland sister clades [39]. Whereas male-male competition has resulted in stark sexual-size dimorphism (larger males) in most primate species, the prevalence of medium sized arboreal predators, particularly the fossa, in Madagascar may have pushed body size selection in both lemur sexes toward the threshold of what the canopy can support. Thus, the anti-predator benefits that larger body size and/or social living provide throughout daily activity cycles may have outweighed the benefits of secure, comfortable sleeping sites such as tree-cavities.

While large body size and social living may relax the need for secure sleep sites, additional factors are necessary to explain the benefits of flexible sleep-site selection over fixed-point nesting; we consider four such factors most important to consider. 1) Proximity to food resources has been shown to influence primate nest-construction and sleep-sites [36, 40, 41]. In landscapes with dynamic resource distributions, flexible sleep-sites could optimize foraging efficiency by reducing daily commutes to resource hotspots that last more than a day. 2) Similarly, sleep-site flexibility may allow adaptive responses to changes in predation risk; why should an animal invest in sleep-site modification if it is likely to be chased off by a predator at any time? 3) Parasite risk may play multiple roles in shaping sleep site selection. Sleeping areas can become contaminated with parasites, suggesting that exposure to parasites might be a cost of reusing sleep-sites [42]. Conversely, certain sleeping sites can provide parasite-related benefits to primates. In the context of vector-borne diseases, for example, an enclosed site may help to obscure cues that mosquitoes use to locate hosts. Supporting this hypothesis, malaria prevalence decreases in species of New World monkeys that sleep in enclosed microhabitats [43]. 4) Finally, factors such as group size and body mass likely limit the ability of some primate species to use fixed-point nesting habitats and, in general, to obtain consolidated sleep [44].

Only with the emergence of frequent, secure platform construction in ancestral apes has deep, REM heavy sleep architecture (re)-emerged. This sleep pattern is most strong expressed in humans [45], perhaps due to a more recent transition to sleeping on the ground in sentinalized groups permitting even higher quality sleep along the human lineage [46]. In general, mammals with larger encephalization quotients exhibit more REM sleep [4]; cognitive ability likely explains some differences in nesting behavior between apes and other primates. While black and white ruffed lemurs (*Varecia variagata*), for example, will build nests regardless of their ability to observe con-specifics doing so [47], whether and how chimpanzees build nests depends on the nest-building behavior of their social group [48]. Accordingly, site-flexible nest construction is thought to reflect great apes' capacity for social learning and environmental problem solving [49], and is considered by some to be the most pervasive form of material culture in great apes [5, 15, 17]. Thus, our results provide further evidence that an integration of the sleep quality hypothesis, sleep-site flexibility, and environmental cognition are necessary to explain early hominid evolution [44]; great apes are not the only primates that would benefit from flexibly and regularly building secure sleep sites, but they are the only ones that do so. One remaining question is whether inter-species cognitive differences also mediate the fitness benefits of sleep quality. For example, primate brain size and diet are correlated, with larger

brained species being more reliant on resources that are predictable, but sparse and ephemeral [50, 51]; does high quality sleep aid species that rely on sophisticated cognitive maps to keep track of temporal patterns of resource availability at multiple locations?

The precise nature of the tradeoffs in sleep-site selection–and the ecological conditions that favor the evolution of one strategy over another–would be further elucidated by expanding the comparisons in sleep among lemur species and between other primates. Though it was our hope that this study might help differentiate the ecological factors mediating the impact of sleep quality, the nearly negligible amount of variance in our data that can be attributed to subject species tells us there is little to be gained from attempting to explicitly model species level effects with the data at hand. This is perhaps unsurprising given the limited number of individuals per species in our study, and speaks more to a lack of statistical power than the absence of species level variation. Our small sample size also prevents us from conducting additional tests to explore other individual-level predictors of TST, such as age or sex. Here, we were limited by the constraints of working with globally threatened species and the logistical difficulty of maintaining controlled sleep conditions in a large, multipurpose facility.

None-the-less, we can offer some insights to researchers who wish to answer these questions through the collection of additional data. We did not, for example, consider in our experimental design the important interaction of experimental condition with the order of presentation. The potential of carryover effects (the impact of one condition differentially affecting sleep during the next condition) resulting from this interaction required us to include additional parameters in our statistical model. Allowing a recovery period between experimental conditions would remove the need to statistically control for this effect, increasing the statistical power per observation without creating additional work or disturbance to the lemurs [32]. We also take a fairly simple approach to modeling the temporal dynamics of lemur sleep due to the limited duration of our observation periods. Study of sleep patterns during extended baseline conditions would allow for more sophisticated models of baseline sleep behavior (e.g. by adding additional moving average or autoregressive parameters to account for long term homeostatic trends). These additional data would not require the time intensive and disruptive efforts of experimental manipulations, but would facilitate inference about the effects of such experiments when they are conducted.

Future work that employs these ideas may also identify other ecological co-variates of increased sleep duration in enriched sleep sites. The inclusion of smaller, fixed-point nesting species in a similar study, for example, could reveal trade-offs between more flexible sleep locations and periods of intense, high-quality sleep in primates. Comparison of lemurs to lorisiforms may also prove fruitful; a comparative analysis of lorisiform sleep behavior revealed predation pressure strongly influenced sleep-site selection, but neither the effect of predation risk nor of specific sleep sites on sleep quality has been investigated [52]. Many lemurs, including those in our study, also display a high degree of cathemerality [38]; with sufficient data, it may be possible to link flexible sleep site locations with flexible sleep timing. As affordable and mobile sleep monitoring technology improves upon current methods, future research can increase the number of subjects per species used in such a study. Though we are limited in the inferences we can make about species level differences in the relationship between sleep site and sleep duration, our experimental approach has clearly demonstrated that there is a relationship in at least some lemurs. With further replication and refinement, this approach can be used to understand not just where and how primates sleep, but why they do so.

In conclusion, we found that enriched sleep-sites increase sleep duration in at least some lemurs. This highlights the importance of sleep-site conditions for lemurs, as is also known for hominoids [12, 53–55] and cercopithecoids [56]. With respect to captive primate welfare, previous work has illustrated the importance of enriching sleep-sites for large brained and large

bodied great apes [15, 57]. In light of our findings, we suggest that managing institutions for any primate species should take care to allot resources to ensure that primates that are primed to sleep in species-specific ways by way of environmentally modified sleep-sites. Finally, we conclude that behaviors that influence sleep-site selection, thereby augmenting sleep quality, are evolutionarily conserved in primates and may be critically important for not only apes and monkeys, but certain strepsirrhines as well.

## Supporting information

**S1 File. Statistical analysis.** Code and diagnostic figures from which the statistical analysis in this paper can be fully replicated.
(DOCX)

## Acknowledgments

We are grateful to the staff at the DLC and offer thanks to Erin Ehmke and David Brewer for continuous support through all aspects of this research. We thank Mark Grote for his statistical consultation and Barbara Fruth for her helpful review of this manuscript. Additionally, we would like to thank the reviewer comments that helped improve previous versions of the manuscript. This research was supported by Duke University. This is Duke Lemur Center publication #1496.

## Author Contributions

**Conceptualization:** Charles L. Nunn, David R. Samson.

**Data curation:** David R. Samson.

**Formal analysis:** Alexander Q. Vining, Charles L. Nunn, David R. Samson.

**Funding acquisition:** Charles L. Nunn.

**Investigation:** Alexander Q. Vining.

**Methodology:** Charles L. Nunn, David R. Samson.

**Project administration:** Alexander Q. Vining, Charles L. Nunn.

**Resources:** Charles L. Nunn.

**Supervision:** Charles L. Nunn.

**Validation:** Charles L. Nunn, David R. Samson.

**Visualization:** Alexander Q. Vining.

**Writing – original draft:** Alexander Q. Vining, David R. Samson.

**Writing – review & editing:** Alexander Q. Vining, Charles L. Nunn, David R. Samson.

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
