## [Decision Letter · Decision Letter 0]

18 Jun 2021

PONE-D-21-17696

Enriched sleep environments lengthen lemur sleep duration

PLOS ONE

Dear Dr. Vining,

Thank you for submitting your manuscript to PLOS ONE. After careful consideration, we feel that it has merit but does not fully meet PLOS ONE’s publication criteria as it currently stands. Therefore, we invite you to submit a revised version of the manuscript that addresses the points raised during the review process.

While the 2nd and 3rd reviewers are overall positive, the 1st reviewer raises several important concerns about the experimental design, as well as data analysis, presentation and interpretation, that require attention.   

We look forward to receiving your revised manuscript.

Kind regards,

Vladyslav Vyazovskiy, PhD

Academic Editor

PLOS ONE

Journal Requirements:

Reviewers' comments:

Reviewer's Responses to Questions

**Comments to the Author**

1. Is the manuscript technically sound, and do the data support the conclusions?

Reviewer #1: No

Reviewer #2: Yes

Reviewer #3: Yes

2. Has the statistical analysis been performed appropriately and rigorously? 

Reviewer #1: No

Reviewer #2: Yes

Reviewer #3: Yes

3. Have the authors made all data underlying the findings in their manuscript fully available?

Reviewer #1: No

Reviewer #2: No

Reviewer #3: Yes

4. Is the manuscript presented in an intelligible fashion and written in standard English?

Reviewer #1: Yes

Reviewer #2: Yes

Reviewer #3: Yes

5. Review Comments to the Author

Reviewer #1: This study addresses the question how sleep environment influence sleep duration in primates. Effects of an enriched or impoverished sleep-site on sleep time were measured in a total of 8 individuals of 4 different lemur species by means of actigraphy and video recordings. The results suggest that an enriched sleep-site, particularly an increased softness and insulation, increases daily sleep time by about half an hour. The study is interesting and well introduced, but I do have a number of questions and concerns regarding the experimental design and data analysis that need to be addressed.

1. In general, there is a lack of detail on the processing and analysis of the data, which in the end makes it hard to appreciate and interpret the value of the finding. It may be a cliché, but recordings of rest-activity patterns are only a correlate of true sleep-wake patterns. How strong the correlation is depends on many variables and may differ between species, sexes, and experimental conditions. More details are needed on how the actigraphy and video data were processed. Particularly, it needs to be explained in detail how these data were then transformed and used as an indicator of sleep time. I also don’t quite understand the operational definition for sleep being “the absence of any force in any direction during the measuring period” (line 119-120).

2. It is somewhat surprising that the manuscript only presents data on total daily resting time whereas the actigraphy and video data surely contain a lot more interesting and relevant information. For example, what about the duration of sleep/rest episodes in the different conditions as a possible measure of sleep continuity? And what about the distribution of sleep/rest and activity across the 24h cycle?

3. The experimental design is complicated. The study aimed to compare actigraphy-based total daily sleep time under baseline conditions with sleep time in an enriched or impoverished nest-site condition. However, the methods section mentions that the Lemurs had access to all the cage enrichments during the day (line 163). This is where the results become difficult to interpret. For example, is sleep in the impoverished condition determined by the impoverished sleep-site at night or by the enriched environment during the day?

4. Another complication in the design is that animals were sleeping in pairs during baseline but not during the enriched and impoverished nest-site conditions (line 135-136 and 143-144). So, how much of the effect that is observed is due to the difference in cage enrichment or simply the fact that the animals where individually housed?

5. Also, if I read the methods section correctly, the baseline condition always preceded the enriched and impoverished conditions. How do the authors exclude the possibility of an order effect or time effect? Do they have additional baseline recordings after the experimental treatment weeks to assess whether sleep time normalized to baseline values?

6. It is not clear why the authors chose to study 4 different species of lemur (with only 2 individuals for each species) rather than 8 individual of a single species. Also, it seems like species was not included as a variable in the analysis.

7. For each of the 4 species, 2 individuals were included, one male and one female. Were there any sex differences in rest-activity patterns?

8. The Figure shows the data in its most basic form; that is, daily resting time separately for each individual on each of the 6-7day in each of the 3 conditions. However, this makes it quite hard to read the figure and get an overall picture. Perhaps the authors should include additional figures or panels that simply shows the average resting times per condition. After all, they claim that the claim is that there is a statistical significant effect of experimental condition.

9. Based on the current figure, the reader may have the impression that the differences in daily resting time between the conditions is rather small compared to even the variation within an individual and within a certain condition. Moreover, the differences between the conditions is certainly small compared to the differences between the individuals. This raises questions as to how relevant these differences are and what other variables might explain variation in resting time. Perhaps this should be discussed.

10. The discussion speaks about the preference of the subjects for enriched sleeping-sited (line 283). However, in this study the lemurs were not offered a choice between the different conditions, so how can we know what their preference was.

11. The discussion mentions that the lemurs chose to sleep on narrow, elevated ledges rather than the ground during the impoverishment condition (line 286). If this is a meaningful observation worthy of discussion, the authors should consider showing the data to support this. In a way this goes back to point 2, the feeling that there is much more in the actigraphy data and video recordings than only the total daily resting time now presented.

Minor comments:

- Methods, line 111: animals were housed indoors to control for temperature and light. Please add what the indoor temperature and light conditions were during the study

- Results, Figure 2: Specify the unit of the y-axis (minutes).

Reviewer #2: Dear Dr Vyazovskiy:

It was my pleasure to read the paper by Vining et al. entitled, “Enriched sleep environments lengthen lemur sleep duration” for the journal PLOS ONE. The paper reports original findings on the effect of sleep site comfort on sleep duration in four species of primate. Although the low size is very low (two animals per species), the data collected is unique and hard to replicate. Moreover, owing to strict regulations on primate research (even behavioural studies), it is best to take whatever data can be had. I recommend the data be published in PLOS ONE subject to minor revision of the paper, as per below.

Title page: Please give City, Country for all affiliations.

Introduction, lines 64, 65: Please define “deep” and “efficient” and “higher quality” sleep. These are never actually explained in the paper, yet much seems to rest on their definition. For instance, sleep depth can be measured directly, with arousal thresholds, or inferred using slow wave activity during slow wave sleep. Slow wave activity cannot be compared across species (for a variety of physiological and non-physiological reasons) and so some explanation would be warranted.

Introduction, line 70. Consider this: The reduced skeletal muscle tone that accompanies REM sleep might make small surface areas particularly dangerous and thus animals sleeping in such locations might selectively reduce REM sleep (either in duration or %sleep).

Methods, line 110: How do you know the animals acclimated to the collars within 2 hours?

Methods: line 115: Total sleep time (TST) is a standard acronym for 24-h sleep amount. Otherwise it is TRT - total recording time. I would suggest you use TST and not TTST as the latter is not familiar to most sleep readers.

Methods, line 123: Are you certain you validated sleep states? First, no data on sleep states is presented. Second, this seems rather a big deal to gloss over. Please explain, clarify or correct.

Methods: the low sample size needs to be addressed. I can guess as to the reasons and am empathetic but it will be eye-catching to many. I’d be proactive and defend against criticism straight-up.

Methods, line 170: Should species not be a fixed factor in the model?

Results, line 223-225: I would not say “sample size of 7”. This is misleading. I would replace with “All values were an average over 7 days, except….”

Discussion, line 255: The issue of sleep depth and quality re-emerges here too. Sleep depth is difficult to compare across species. You cannot compare SWA across species. Moreover, some data challenges comparing arousal thresholds across species. For instance, sleeping emperor penguins arouse quickly with lightly touched on their feet, but no where else on their body. Does this make an emperor penguin a light or deep sleeper compared to a starling? I have strong doubts over the ability to make such statements.

Discussion, line 295-297: This result was also found by Lesku et al. 2006 in that mammalian species with greater encephalization had a higher %REM sleep. Thus, not only do humans have more REM sleep (%) than non-human primates but so too do mammals, in general.

References: There are heaps of sloppy errors in the reference section. Endnote appears to have been used, but unchecked. Please check each reference individually for consistent formatting and accuracy.

Reviewer #3: First, I would like to say that the ethics statement in this study was excellent and reassuring - when I read "sleep impoverishment" alarm bells rang for me, although I know that at DLC, all care would be given to the animals, and the excellent and detailed ethics statement made that clear.

Overall, this is a "neat" study with an excellent design. It is novel and indeed, for me, not only has implications for primate evolution and nest building, but my first thought went to conservation, where habitats are destroyed and nesting materials may not be available, long loss of even half an hours sleep per night could have long term negative effects. I think the authors should add a few lines about this to the discussion and perhaps one phrase in the abstract.

Introduction

Line 72 - not all apes build nests - I guess you mean great apes - it would be nice to have a such as after sleep site selection behaviours with a few examples as this paragraph reads very thin

72-75 - unpack this sentence - need a better flow into lemurs - maybe that many strepsirrhines build nests, and a more detailed review of the main functions (also comfort, parasite control etc)

77 - I would be more broad since most galagos also use nests - and many lemurs don't - I feel this work has broader impact if you are looking at strepsirrhine evolution - there are a few good reviews on sleep site use by galagos (Bearder earlier on, Svensson more recently) - then justify the choice of the lemur species selected...I actually had no idea that sifakas use nests...nor ring-tailed lemurs - this makes me realise that the previous paragraph is confusing - are you testing just sleep sites? or nests? clarify and justify the examples

Line 86 - I feel that this section should be either be higher or in the discussion - you can simply state that you discuss sleep site comfort in relation to welfare and evolution (and I suggest add conservation)

Methods - really cool experimental design! I am just curious about battery life and if you had to change the collars or recharge the battery (or remove the sensor regularly for download of data)

Discussion - AH now you say these are species that do NOT nest - this needs to be more clear in the intro

Line 241 - can you not compare with any other studies? You mentioned a mouse lemur study in the wild...you mentioned several bird studies - and surely some human studies - no references in this paragraph

Line 261 - as this is not a primate journal, you need to explain what cathemeral is

Line 263 - turn this around to explain to the reader about sex dimorphism then you can interpret the findings

Lie 268 - the lemurs in your sample certainly do not fit the definition of large body size in a mammal - but medium bodied - Maybe you want to include a threshold in grams related to security in branches to support a size of that nature - this is why virtually no large bodied mammals are arboreal (the exception mainly being some great apes!)

Paragraph 268 - a lot to unpack in this paragraph - follow the hypotheses paragraph by paragraph rather than mixing

Line 297 - this implies that human evolved to sleep in tree platforms - do you think this is the case? provide more evidence

Line 300 - this feels a bit 1950s and very primatological - most animals that make nests have some form of learning, at least in choice of materials etc

Line 305 - do you mean hominid?? or do you mean hominin? are you including the ancestors of chimps etc?

Line 309 - bold statement - reference needed

Line 319- globally threatened - not endangered (that is an IUCN category)

310-335 - a lot of unreferenced reflection - can be dramatically shortened or made stronger with referencing

Suggest here to include something about conservation implications

Line 355 - substandard is an odd choice- indeed, captive facilities should consider more the ecology of the species and try to closely match it

Line 357 - larger bodied apes and monkeys (again monkeys are not large for the most part)...strepsirrhines unless you mean tarsiers but you never mentioned tarsiers anywhere else and you also did not mention anything other than lemurs so this comes out of the blue anyway! Indeed, there are some very nice new studies of sleep in slow lorises using similar methods as to here, as well as the aforementioned studies on galagos that would really complement the work and make the discussion broader

Some minor comments

Line 65 etc - the accepted spelling of orang-utan is generally with a hyphen and at the very least this is how Indonesians would spell it in English

Figure 1 text a subject (Propithecus coquereli)

Table 1 - random change of font and the table heading needs to be more detailed and explain the parameters

line 235 - among the models THAT we compared

Line 274 - delete you - replace with alternative - an animal, a lemur, a primate

6. PLOS authors have the option to publish the peer review history of their article (what does this mean?). If published, this will include your full peer review and any attached files.

Reviewer #1: No

Reviewer #2: No

Reviewer #3: No

---

## [Author Response · Author response to Decision Letter 0]

24 Aug 2021

Reviewer 1

1. In general, there is a lack of detail on the processing and analysis of the data, which in the end makes it hard to appreciate and interpret the value of the finding. It may be a cliché, but recordings of rest-activity patterns are only a correlate of true sleep-wake patterns. How strong the correlation is depends on many variables and may differ between species, sexes, and experimental conditions. More details are needed on how the actigraphy and video data were processed. Particularly, it needs to be explained in detail how these data were then transformed and used as an indicator of sleep time. I also don’t quite understand the operational definition for sleep being “the absence of any force in any direction during the measuring period” (line 119-120). 

We are thankful to the reviewer for this input. We added the raw actigraphy data collected from our collars to the public repository for this publication (https://github.com/aqvining/Lemur_Sleep_Site_Enrichment) and the code to calculate TST and all other secondary metrics to Supplementary Material 1. Additinally, upon reviewing this process, the initial steps of our data processing were not fully replicable, and we found an error with the TST calculations for one individual. We corrected this, which led to some minor changes to the quantitative results and no changes to conclusions. 

Unfortunately, CamNtech has been unwilling to share the algorithm by which their actigraphy counts (our raw data) are derived from underlying acceleration measurements. However, we added citation to a paper that attempts to recreate this algorithm for a similar collar and also clarified the relationship between actigraphy counts and our inference of sleep state (lines 158-163). 

Data from infrared videography collected in the course of the current study were not collected in a reproducible manner, so we have removed reference to those data. Instead, we discuss how infrared videos were used to more rigorously validate our protocol for inferring sleep state from CamNtech MotionWatch8 collars in a previous study at the Duke Lemur Center (lines 165-170)

2. It is somewhat surprising that the manuscript only presents data on total daily resting time whereas the actigraphy and video data surely contain a lot more interesting and relevant information. For example, what about the duration of sleep/rest episodes in the different conditions as a possible measure of sleep continuity? And what about the distribution of sleep/rest and activity across the 24h cycle? 

Thank you for these suggestions. We made the raw actigraphy data available, as described in our response to the previous comment. We also added analyses to the paper to investigate the inter-daily stability and intra-daily variation of actigraphy data. New predictions regarding these metrics can be found in lines 114-116, the calculation and analysis of these metrics are now provided in lines 268-283, results from these analyses, including a new figure, are in lines 319-336, and we discuss these new results in lines 365-375. We have also modified Table 2 to present these data, removing standard deviations of TST to keep the data presentation compact and readable.

3. The experimental design is complicated. The study aimed to compare actigraphy-based total daily sleep time under baseline conditions with sleep time in an enriched or impoverished nest-site condition. However, the methods section mentions that the

Lemurs had access to all the cage enrichments during the day (line 163). This is where the results become difficult to interpret. For example, is sleep in the impoverished condition determined by the impoverished sleep-site at night or by the enriched environment during the day? 

Thank you for this comment. In the referenced line, we confused the “enrichment condition” specific to our study with the assortment of items added to lemur enclosures to create a more dynamic, habitat-relevant environment, often also referred to as enrichment outside the context of this paper. We rephrased to remove this ambiguity.

4. Another complication in the design is that animals were sleeping in pairs during baseline but not during the enriched and impoverished nest-site conditions (line 135-136 and 143-144). So, how much of the effect that is observed is due to the difference in cage enrichment or simply the fact that the animals where individually housed? 

We appreciate the reviewer’s attention to possible confounds in our experimental design. Because the social aspect of sleep conditions is controlled for in the contrast of our enriched vs. impoverished conditions, we can safely conclude isolated sleep does not explain the observed differences in total sleep time between these two conditions. This is noted in lines 345-346 and we have now further clarified our rationale for the decision in lines 186-190 so that it is clear to the reader when we introduce our methods. Though we successfully control for the confound of social sleeping in the contrast between enriched and impoverished conditions, we agree that this confound would affect the absolute measures of difference from baseline. This raises the possibility that our results could potentially be explained by an increase of TST from sleeping alone PLUS a decrease in TST from the impoverished condition, rather than simply an increase in TST during the enriched condition. We have added this caveat to our conclusions (line 361).

5. Also, if I read the methods section correctly, the baseline condition always preceded the enriched and impoverished conditions. How do the authors exclude the possibility of an order effect or time effect? Do they have additional baseline recordings after the experimental treatment weeks to assess whether sleep time normalized to baseline values? 

The reviewer is correct that the baseline condition always precedes the enriched and impoverished conditions, and is astute to note the possibility of an order effect. We explicitly account for the possibility that the order of experimental conditions may affect observed TST by including the parameter Order in our statistical model. Additionally, we account for the possibility that the effect of Enrichment preceding Impoverishment may differ from the effect of Impoverishment preceding Enrichment by including an interaction term between Order and Condition in our statistical model. When calculating our observed differences in TST between conditions, we do so marginal to Order (258, 291), meaning we used the estimated coefficients for Order and its interaction with Condition to calculate the differences in TST between conditions attributable to order and crossover effects. We then subtract these estimates from our final, reported values. The details of these calculations (which use matrix multiplication to propagate estimated error in our coefficients through to our final confidence intervals) are included in the Statistical Inference section of S1: Analysis. 

To clarify the use of these methods in our manuscript without becoming overly technical, we added new language and additional citations (238). We have also rephrased line 345 to explicitly link the technical term “marginal contrasts” to the way we controlled for order effects. 

As the reviewer notes, experimental controls such as additional baseline periods would be helpful, potentially adding statistical power to future analyses; we clarify the role the order and carryover effects would have on such decisions in our discussion of future research (lines 460-470).

6. It is not clear why the authors chose to study 4 different species of lemur (with only 2 individuals for each species) rather than 8 individual of a single species. Also, it seems like species was not included as a variable in the analysis. 

We agree with the reviewer that a study including 8 individuals of the same species and sex would allow for more precise conclusions. Unfortunately, we did not have access to any single species at the DLC for which that many pair-housed individuals were available for this type of study. Additionally, a motivating factor for this study was the development and use of standardized methods with which we could compare the results across species. Unfortunately, our limited sample size did not allow us to directly test hypotheses of species level differences. We do, however, include species in our model as a random effect (U). From the Interclass Correlation Coefficients reported in line 297 we concluded that only a small proportion of the unstructured variance in our modeled data were attributable to species level differences; including species as a fixed effect in our model would be unlikely to reveal significant effects while creating a major risk of overfitting the model. We discuss the rationale behind these decisions in detail in lines 447-459. To address the reviewer’s concerns, we have also added clarifying language at the top of our discussion section (lines 346-349)

7. For each of the 4 species, 2 individuals were included, one male and one female. Were there any sex differences in rest activity patterns?

Thank you for the question, it would indeed be interesting to know if there are any sex differences in the sleep of these species. Unfortunately, due to the small number of individuals in our study, we are limited in our ability to test additional hypotheses. Attempting to simultaneously model species, sex, and individual level effects on TST leads to overparameterization and an unidentifiable model. Because we have no specific hypotheses about the role of sex on total sleep time, and attempting separate tests for any such effect would constitute questionable use of multiple testing, we have chosen to refrain from reporting and speculating on sex-biased total sleep time. We added a sentence at line 455 to help explain our rationale.

8. The Figure shows the data in its most basic form; that is, daily resting time separately for each individual on each of the 6- 7 day in each of the 3 conditions. However, this makes it quite hard to read the figure and get an overall picture. Perhaps the authors should include additional figures or panels that simply shows the average resting times per condition. After all, they claim that the claim is that there is a statistical significant effect of experimental condition 

Thank you for the suggestion. We have added an additional figure (Figure 3) that shows the estimated mean and standard error for the difference (in minutes) of total sleep time between conditions.

9. Based on the current figure, the reader may have the impression that the differences in daily resting time between the conditions is rather small compared to even the variation within an individual and within a certain condition. Moreover, the differences between the conditions is certainly small compared to the differences between the individuals. This raises questions as to how relevant these differences are and what other variables might explain variation in resting time. Perhaps this should be discussed. 

Thank you for the comment. We hope that the figure added in response to the previous comment, along with our discussion of the sources of variance in these data in lines 450-453 will be sufficient to address these concerns.

10. The discussion speaks about the preference of the subjects for enriched sleeping-sited (line 283). However, in this study the lemurs were not offered a choice between the different conditions, so how can we know what their preference was. 

The discussion mentions that the lemurs chose to sleep on narrow, elevated ledges rather than the ground during the impoverishment condition (line 286). If this is a meaningful observation worthy of discussion, the authors should consider showing the data to support this. In a way this goes back to point 2, the feeling that there is much more in the actigraphy data and video recordings than only the total daily resting time now presented. 

We appreciate the reviewer’s concern. At question are the observations that lemurs prefer to sleep in enriched sleep-sites crates (relative to the rest of their enclosure) when available, and on narrow ledges (relative to the rest of their enclosure) when no enrichment was available. These observations were made through a mix of the authors’ experiences, reports by DLC caretaking staff, and a limited sub-sample of the data for which infrared video was available. However, as this evidence is neither un-biased nor fully reproducible, we removed the relevant paragraph.

11. Minor Comments: Thank you, we have made changes to address these.

Reviewer 2

1. Introduction, lines 64, 65: Please define “deep” and “efficient” and “higher quality” sleep. These are never actually explained in the paper, yet much seems to rest on their definition. For instance, sleep depth can be measured directly, with arousal thresholds, or inferred using slow wave activity during slow wave sleep. Slow wave activity cannot be compared across species (for a variety of physiological and non-physiological reasons) and so some explanation would be warranted. 

Thank you for the comment, we agree that some additional clarity was needed. We revised lines 67-70 to more accurately define the measures reported in the studies being described.

2. Introduction, line 70. Consider this: The reduced skeletal muscle tone that accompanies REM sleep might make small surface areas particularly dangerous and thus animals sleeping in such locations might selectively reduce REM sleep (either in duration or %sleep). 

A useful note, thank you. We used it to enrich the language at line 73.

3. Methods, line 110: How do you know the animals acclimated to the collars within 2 hours? 

Thank you for your question. We have rephrased to clarify the protocol by which DLC caretaking staff monitored the lemurs to ensure lemurs were not responding adversely to the collars.

4. Methods: line 115: Total sleep time (TST) is a standard acronym for 24-h sleep amount. Otherwise it is TRT - total recording time. I would suggest you use TST and not TTST as the latter is not familiar to most sleep readers.

Thank you for this useful suggestion. We have adopted the use of TST throughout our materials.

5. Methods, line 123: Are you certain you validated sleep states? First, no data on sleep states is presented. Second, this seems rather a big deal to gloss over. Please explain, clarify or correct

Thank you pointing out this ambiguity. We have described our process for inferring sleep state in greater detail in lines 160-170.

6. Methods: the low sample size needs to be addressed. I can guess as to the reasons and am empathetic but it will be eyecatching to many. I’d be proactive and defend against criticism straight-up.

We understand the reviewer’s concern and appreciate the input. To more proactively address the issue of sample size, we added a sentence to the beginning of the Discussion (lines 346-349).

7. Methods, line 170: Should species not be a fixed factor in the model?

Ideally, yes. However, one of the limitations of our sample size is in our ability to add additional parameters to the model. We compared inter-class correlation coefficients to determine when our statistical model had reached a level of parameterization that approached the limits of our ability to make sound inferences – a process we have now clarified in lines 261-262. Unfortunately, given the large ratio of individual-level variance to species-level variance when both were included as random variables, we lack the ability to detect species level effects even should they exist, and thus refrain from further complexifying our model by including species fixed-effect parameters. This is discussed in lines 455-458.

8. Results, line 223-225: I would not say “sample size of 7”. This is misleading. I would replace with “All values were an average over 7 days, except….”

Thank you for the suggestion, we have adopted it.

9. Discussion, line 255: The issue of sleep depth and quality re-emerges here too. Sleep depth is difficult to compare across species. You cannot compare SWA across species. Moreover, some data challenges comparing arousal thresholds across species. For instance, sleeping emperor penguins arouse quickly with lightly touched on their feet, but no where else on their body. Does this make an emperor penguin a light or deep sleeper compared to a starling? I have strong doubts over the ability to make such statements.

Thank you, we agree that is important to be clear about what exactly has been measured and to draw justifiable conclusions. We rephrased this paragraph to focus on sleep duration instead of sleep quality. Additionally, we added analyses to explore stability and fragmentation of sleep patterns during our experiment, detail of which are provided in response to Reviewer 1’s second question.

10. Discussion, line 295-297: This result was also found by Lesku et al. 2006 in that mammalian species with greater encephalization had a higher %REM sleep. Thus, not only do humans have more REM sleep (%) than non-human primates but so too do mammals, in general.

Thank you for highlighting these results here; we have used an additional citation to the study mentioned to strengthen our argument in this paragraph (line 428).

11. References: There are heaps of sloppy errors in the reference section. Endnote appears to have been used, but unchecked. Please check each reference individually for consistent formatting and accuracy.

We thank the review for their attention to detail, and apologize for the mistakes – we failed to realize our reference manager was overwriting manual fixes with automatic updates. We have carefully checked the references upon final submission and made corrections where necessary.

Reviewer 3

1. Overall, this is a "neat" study with an excellent design. It is novel and indeed, for me, not only has implications for primate evolution and nest building, but my first thought went to conservation, where habitats are destroyed and nesting materials may not be available, long loss of even half an hours sleep per night could have long term negative effects. I think the authors should add a few lines about this to the discussion and perhaps one phrase in the abstract.

Thank you for the encouraging words! We share your enthusiasm for conservation and likewise hope that our research might facilitate conservation efforts. However, we have struggled to come up with direct links between our findings and currently employed strategies for conservation management. We would feel uncomfortable promoting the conservation benefits of our study without a better understanding of what these might be, especially given that the subjects in our study are not primarily nest builders.

2. Introduction

Line 72 - not all apes build nests - I guess you mean great apes - it would be nice to have a such as after sleep site selection behaviours with a few examples as this paragraph reads very thin

72-75 - unpack this sentence - need a better flow into lemurs - maybe that many strepsirrhines build nests, and a more detailed review of the main functions (also comfort, parasite control etc)

77 - I would be more broad since most galagos also use nests - and many lemurs don't - I feel this work has broader impact if you are looking at strepsirrhine evolution - there are a few good reviews on sleep site use by galagos (Bearder earlier on,

Svensson more recently) - then justify the choice of the lemur species selected...I actually had no idea that sifakas use nests...nor ring-tailed lemurs - this makes me realise that the previous paragraph is confusing - are you testing just sleep sites? or nests? clarify and justify the examples 

Line 86 - I feel that this section should be either be higher or in the discussion - you can simply state that you discuss sleep site comfort in relation to welfare and evolution (and I suggest add conservation)

. . .

Discussion - AH now you say these are species that do NOT nest - this needs to be more clear in the intro

Thank you for the detailed suggestions for how we can improve our introduction and discussion of sleep phenotype variation in primates. We have approached this set of comments together, substantially expanding and re-organizing the final two paragraphs (lines 78-121) of our introduction to include additional examples and clearer justification for the choice of species in our study.

3. Methods - really cool experimental design! I am just curious about battery life and if you had to change the collars or recharge the battery (or remove the sensor regularly for download of data)

Thank you! The collars stayed on the lemurs for the duration of study with no need to recharge or download data. The battery life of the collars was one of the determining factors in the length of each study period – adding additional baseline periods, for example, would likely have required us to catch the lemurs and replace their collars.

4. Line 241 - can you not compare with any other studies? You mentioned a mouse lemur study in the wild...you mentioned several bird studies - and surely some human studies - no references in this paragraph

The paragraph referenced discusses the specific results of our study and the statistical support behind our conclusions; we do not have any relevant references to add. As the reviewer notes, we discuss these results in the larger context of other studies at different points in our Introduction and Discussion. 

5. Line 261 - as this is not a primate journal, you need to explain what cathemeral is

 Thank you for the observation. We have added clarity to our definition at line 384.

6. Line 263 - turn this around to explain to the reader about sex dimorphism then you can interpret the findings

Lie 268 - the lemurs in your sample certainly do not fit the definition of large body size in a mammal - but medium bodied - Maybe you want to include a threshold in grams related to security in branches to support a size of that nature - this is why virtually no large bodied mammals are arboreal (the exception mainly being some great apes!)

We appreciate the opportunity to improve clarity here, and have added a sentence at line 388 to address both of these points.

7. Paragraph 268 - a lot to unpack in this paragraph - follow the hypotheses paragraph by paragraph rather than mixing.

Thank you for pointing out the density of this paragraph. We have added some organizational phrasing and numbering to help distinguish the hypotheses and make this paragraph easier to digest.

8. Line 297 - this implies that human evolved to sleep in tree platforms - do you think this is the case? provide more evidence

Thank you for this observation. Out intent was to imply that common ancestors of apes and humans evolved to sleep in tree platforms, but that early hominins likely transitioned to different sleep patterns sometime after their divergence from other Great Apes. We rewrote these two sentences (lines 425-427) to clarify.

9. Line 300 - this feels a bit 1950s and very primatological - most animals that make nests have some form of learning, at least in choice of materials etc.

We appreciate the note, and have rephrased to clarify the results of studies being cited and add nuance to distinguishing features of learning being discussed.

10. Line 305 - do you mean hominid?? or do you mean hominin? are you including the ancestors of chimps etc?

Correct, we mean to use hominid. In this paragraph, we discuss the emergence of secure, flexible platform building in ancestral apes, and how our research helps us understand the context of this evolutionary shift. We have elaborated on the sentence at line 439 to help reinforce this point.

11. Line 309 - bold statement - reference needed

We have rephrased to make it clearer this was meant as a question, not a statement, and added reference to some of the underlying literature that motivated the question (line 442).

12. Line 319- globally threatened - not endangered (that is an IUCN category)

Thank you, we corrected this.

13. 310-335 - a lot of unreferenced reflection - can be dramatically shortened or made stronger with referencing

We added a reference to back up some statistical suggestions made in this section. Largely, however, we are leaning on our own expertise and experiences conducting this study to make suggestions for improving similar, future efforts. We believe that a thorough discussion regarding the strengths and weaknesses of a study such as the one presented here belongs in any manuscript. As the subject of this discussion is our own study and what we learned in the process of conducting it, we do not have additional references to add. 

14. Line 355 - substandard is an odd choice- indeed, captive facilities should consider more the ecology of the species and try to closely match it

Thank you. We agree that the ending clause containing “substandard” distracts from this point and we removed it.

15. Line 357 - larger bodied apes and monkeys (again monkeys are not large for the most part)...strepsirrhines unless you mean tarsiers but you never mentioned tarsiers anywhere else and you also did not mention anything other than lemurs so this comes out of the blue anyway! Indeed, there are some very nice new studies of sleep in slow lorises using similar methods as to here, as well as the aforementioned studies on galagos that would really complement the work and make the discussion broader

Thank you for helping us to add clarity here. We have removed “large bodied” to avoid ambiguity, changed “prosimian” to strepsirrhine (line 500), and added reference to Svensson et al.’s comparative study of lorisiform sleep at line 478.

Some minor comments

Line 65 etc - the accepted spelling of orang-utan is generally with a hyphen and at the very least this is how Indonesians would spell it in English

Figure 1 text a subject (Propithecus coquereli)

Table 1 - random change of font and the table heading needs to be more detailed and explain the parameters

line 235 - among the models THAT we compared

Line 274 - delete you - replace with alternative - an animal, a lemur, a primate

Thank you, we have made the suggested changes.

---

## [Decision Letter · Decision Letter 1]

27 Sep 2021

PONE-D-21-17696R1Enriched sleep environments lengthen lemur sleep durationPLOS ONE

Dear Dr. Vining,

Thank you for submitting your manuscript to PLOS ONE. After careful consideration, we feel that it has merit but does not fully meet PLOS ONE’s publication criteria as it currently stands. Therefore, we invite you to submit a revised version of the manuscript that addresses the remaining minor points raised during the review process. Please submit your revised manuscript by Nov 11 2021 11:59PM. If you will need more time than this to complete your revisions, please reply to this message or contact the journal office at plosone@plos.org. Please include the following items when submitting your revised manuscript:A rebuttal letter that responds to each point raised by the academic editor and reviewer(s). You should upload this letter as a separate file labeled 'Response to Reviewers'.A marked-up copy of your manuscript that highlights changes made to the original version. You should upload this as a separate file labeled 'Revised Manuscript with Track Changes'.An unmarked version of your revised paper without tracked changes. You should upload this as a separate file labeled 'Manuscript'.If applicable, we recommend that you deposit your laboratory protocols in protocols.io to enhance the reproducibility of your results. Protocols.io assigns your protocol its own identifier (DOI) so that it can be cited independently in the future. For instructions see: https://journals.plos.org/plosone/s/submission-guidelines#loc-laboratory-protocols. Additionally, PLOS ONE offers an option for publishing peer-reviewed Lab Protocol articles, which describe protocols hosted on protocols.io. Read more information on sharing protocols at https://plos.org/protocols?utm_medium=editorial-email&utm_source=authorletters&utm_campaign=protocols.

We look forward to receiving your revised manuscript.

Kind regards,

Vladyslav Vyazovskiy, PhD

Academic Editor

PLOS ONE

Journal Requirements:

Reviewers' comments:

Reviewer's Responses to Questions

**Comments to the Author**

1. If the authors have adequately addressed your comments raised in a previous round of review and you feel that this manuscript is now acceptable for publication, you may indicate that here to bypass the “Comments to the Author” section, enter your conflict of interest statement in the “Confidential to Editor” section, and submit your "Accept" recommendation.

Reviewer #1: (No Response)

Reviewer #2: All comments have been addressed

2. Is the manuscript technically sound, and do the data support the conclusions?

Reviewer #1: Partly

Reviewer #2: Yes

3. Has the statistical analysis been performed appropriately and rigorously? 

Reviewer #1: Yes

Reviewer #2: Yes

4. Have the authors made all data underlying the findings in their manuscript fully available?

Reviewer #1: Yes

Reviewer #2: Yes

5. Is the manuscript presented in an intelligible fashion and written in standard English?

Reviewer #1: Yes

Reviewer #2: Yes

6. Review Comments to the Author

Reviewer #1: The authors did an admirable job in addressing the comments and the revision is certainly a major improvement. While some aspects of the experimental design remain a potential confound (e.g., previous points 4 and 5), at least now the authors acknowledge and address these issues in the discussion.

One issue that has not been fully addressed yet is that recordings of rest-activity patterns are only a correlate of true sleep-wake patterns (mentioned under my previous point 1). In other words, the authors equate a lack of movement as measured with motion watches with sleep. However, it is not excluded that differences in the the amount of movement between conditions reflect, perhaps partly, differences in relaxed wakefulness rather than differences in true sleep. The lemurs in the enriched condition with the high quality sleeping-box, lined with comfortable soft bedding may just be more inclined to stay in their nest a bit longer, even when they are awake. I feel this is not a trivial point, given that the differences in rest and/or sleep between the enriched and impoverished condition amounts to not much more than 30 min on average.

Also, with the previous points in mind, I have mild concerns about the use of the wording 'strong evidence' in both the abstract (line 37) and discussion (line 316). They may want to tone down a little bit.

Reviewer #2: I am satisfied that the authors have effectively addressed my comments and queries and thank them for their thoughtful responses.

7. PLOS authors have the option to publish the peer review history of their article (what does this mean?). If published, this will include your full peer review and any attached files.

Reviewer #1: No

Reviewer #2: No

---

## [Author Response · Author response to Decision Letter 1]

4 Oct 2021

1. “One issue that has not been fully addressed yet is that recordings of rest-activity patterns are only a correlate of true sleep-wake patterns (mentioned under my previous point 1). In other words, the authors equate a lack of movement as measured with motion watches with sleep. However, it is not excluded that differences in the the amount of movement between conditions reflect, perhaps partly, differences in relaxed wakefulness rather than differences in true sleep. The lemurs in the enriched condition with the high quality sleeping-box, lined with comfortable soft bedding may just be more inclined to stay in their nest a bit longer, even when they are awake. I feel this is not a trivial point, given that the differences in rest and/or sleep between the enriched and impoverished condition amounts to not much more than 30 min on average.”

We would like to thank the reviewer for their concern, and agree that we should be as transparent as possible about what we measured. Though we think that our inferential criteria for categorizing sleep are quite conservative and unlikely to capture much restful wakefulness, we have added a few sentences to make clear to our readers that it is a possibility. First, we added a sentence to the methods (line 157) explaining the possibility that we may be capturing wakefulness in our quantitative definition of sleep. Additionally, we now highlight in the first paragraph of our discussion (line 228) that using body-motion as a proxy for sleep limits our ability to make conclusions about specific sleep states such as REM of SWS.

2. Also, with the previous points in mind, I have mild concerns about the use of the wording 'strong evidence' in both the abstract (line 37) and discussion (line 316). They may want to tone down a little bit.

Thank you for the note. We removed “strong” from both of these sentences.

---

## [Editor Report · Decision Letter 2]

5 Oct 2021

Enriched sleep environments lengthen lemur sleep duration

PONE-D-21-17696R2

Dear Dr. Vining,

We’re pleased to inform you that your manuscript has been judged scientifically suitable for publication and will be formally accepted for publication once it meets all outstanding technical requirements.

Kind regards,

Vladyslav Vyazovskiy, PhD

Academic Editor

PLOS ONE
---

## [Editor Report · Acceptance letter]

11 Oct 2021

PONE-D-21-17696R2 

Enriched sleep environments lengthen lemur sleep duration 

Dear Dr. Vining:

I'm pleased to inform you that your manuscript has been deemed suitable for publication in PLOS ONE. Congratulations! Your manuscript is now with our production department. 

Kind regards, 

on behalf of

Dr. Vladyslav Vyazovskiy 

Academic Editor

PLOS ONE